# Knowledge and attitudes about conduct disorder of professionals working with young people: The influence of occupation and direct and indirect experience

Chloe Pinchess[1,2]*, Ruth Pauli[3], John Tully[4,5]

1 Centre for Forensic and Family Psychology, School of Medicine, University of Nottingham, Nottingham, United Kingdom, 2 Clayfields House, Nottinghamshire County Council, Nottingham, United Kingdom, 3 Centre for Human Brain Health, School of Psychology, University of Birmingham, Birmingham, United Kingdom, 4 Academic Unit of Mental Health and Clinical Neurosciences, School of Medicine, University of Nottingham, Nottingham, United Kingdom, 5 Nottinghamshire Healthcare NHS Trust, Nottingham, United Kingdom

* chloepinchess@icloud.com

## Abstract

### Background

Knowledge and attitudes of professionals both pose a potential barrier to diagnosis and treatment of mental disorders. However, knowledge and attitudes about conduct disorder in professionals working with young people are poorly understood. Little is known about the impact of occupation, direct and indirect (training and education) experience, or the interrelationship between knowledge and attitudes.

### Methods

We conducted an online survey of 58 participants, including Psychology Staff, Teaching Staff, Care Staff, and Other Non-Clinical Staff. A questionnaire comprising three subscales (causes, treatments, and characteristics) measured knowledge. A thermometer scale measured global attitudes. Open-ended response measures were used to measure four attitude components: stereotypic beliefs (about characteristics), symbolic beliefs (about the holder's traditions), affect, and past behaviour. Primary analysis explored the impact of occupation, direct experience, and indirect experience on outcome measures. A secondary exploratory analysis was conducted to explore the relationship between knowledge and attitudes.

### Results

Psychology Staff had significantly more favourable global attitudes ($F = 0.49$, $p = 0.01$) and symbolic beliefs ($F = 0.57$, $p = 0.02$) towards those with conduct disorder than Teaching Staff; there were no other significant group differences in attitudes. Psychology staff had more knowledge about conduct disorder than other groups, though the differences were not significant. Direct and indirect experience were associated with greater knowledge (direct: d = 0.97, p = 0.002; indirect d = 0.86, p = 0.004) and favourable global attitudes (direct: d =

**Data Availability Statement:** Data is available at http://doi.org/10.17639/nott.7331. Email addresses and demographic data have been omitted, as

stated in the data sharing section of the consent form (ethics reference FMHS 189-0221).

**Funding:** The only funding provided by the University of Nottingham was to CP for the prize draw vouchers, with a total reimbursement value of £100. The University of Nottingham played no role in study design, data collection and analysis, decision to publish, or preparation of manuscript. The contributing authors have no financial relationships to disclose.

**Competing interests:** The authors have declared that no competing interests exist.

**Abbreviations:** ADHD, Attention deficit hyperactivity disorder; CD, Conduct disorder; CU, Callous un-emotional.

1.12, $p < 0.001$; indirect: $d = 0.68$, $p = 0.02$). Secondary exploratory analyses revealed significant positive correlations between: all knowledge variables with global attitudes; total knowledge with past behaviour; and affect and knowledge of causes with past behaviour.

## Conclusions

Psychology-based staff may have more favourable attitudes towards children with conduct disorder than teachers, primarily due to direct and indirect experience with the disorder. Our sample may have been too small to detect overall or within-group effects of knowledge or attitudes, however exploratory analyses showing a positive correlation between knowledge and attitudes suggest education may be critical in supporting teachers and other groups in their approaches to this challenging group of young people.

## Background

Conduct disorder (CD) is a childhood behaviour disorder characterised by persistent disregard for others and a repetitive pattern of antisocial behaviour including theft, lies, and physical violence [1]. Also, following extensive debate in scientific literature about the importance of callous-unemotional (CU) traits, DSM-5 includes the specifier with or without 'limited prosocial emotions', such as lack of empathy, akin to CU traits [1]. Estimates suggest that 2–2.5% of individuals worldwide have CD [2], with consistency across geographical regions [3]. The diagnosis is approximately twice as common in males compared to females [2, 3].

Conduct disorder has considerable societal and personal consequences, primarily due to the strong associations between CD and violence [4] and life-course persistent offending [5]. It is associated with a substantial economic burden, with an estimated cost of £5,569 per child with CD over three years. Costs include care, mental health, and educational services [6], however between 19% and 64% of these total costs are accounted for by criminal justice services [7, 8]. Financial costs extend into adulthood, with around 50% of those with CD going on to develop antisocial personality disorder [9]. Furthermore, CD is associated with the development of other mental disorders in adolescence and later life, including anxiety disorders and depression [10], as well as particularly high rates of substance misuse [4, 11], which places significant strain on mental health services and reduces quality of life for the individual. Additionally, CD is associated with lower educational attainment [4], which is independently associated with higher levels of stress, negative health consequences, and poorer long-term job prospects [12].

Despite these serious implications, CD is suboptimally managed in healthcare settings. Various professionals work with children with CD, including teachers, psychologists, and care workers [13]. However, it often goes undiagnosed, perhaps due to a prejudice that CD is not a "legitimate" psychiatric disorder [14]. Also, clinical application of the evidence base for the management of CD remains limited. Although studies exploring the efficacy of both pharmacological and psychosocial interventions yield at best mixed findings, there is some evidence for effective treatments. For instance, meta-analyses demonstrate that psychosocial interventions such as multi-systemic therapy produce small effects on offending, psychopathology, and substance use in young people, showing most efficacy in youth under 15 years old [15, 16]. Also, antipsychotics and psychostimulants can reduce symptoms of aggression in CD, albeit possibly only in the short-term [17, 18]. Despite these therapeutic options, only a quarter of

children with CD access specialist health services, possibly due to more practical support being sought rather than mental health assessment and treatment [19]. Furthermore, children with CD may be likely to drop out of treatment because it may not match the needs of the family [20].

One barrier to successful management of CD may be professionals' lack of knowledge. This may be due to limited understanding about the benefits of treatments, meaning professionals may fail to make necessary referrals [21]. Knowledge of mental illness increases alongside experience, leading to reduced stigma and more favourable behaviours towards patients [22]. However, evidence for knowledge specifically about CD in staff working with youth populations is limited, with a particular lack of quantitative data. Some qualitative studies suggest that teachers have little knowledge of CD characteristics and management techniques [23–25]. Other data suggest that psychiatrists have varied knowledge of the aetiologies of CD [26]. While no study to date has explored the impact of knowledge specifically on the management of CD, studies in young people with attention deficit hyperactivity disorder (ADHD) have demonstrated that teachers with greater knowledge more readily refer children for assessment due to a greater understanding of the challenges they face [21]. Similarly, research shows that teachers with greater experience have more favourable behaviours towards children with ADHD [27].

Staff attitudes are another factor that may influence the management and outcomes of CD. Studies in adults with mental disorders demonstrate that negative staff attitudes towards patients results in avoidance of patients [28] and lower patient satisfaction [29]. In contrast, healthcare workers who have experience of working with those with mental illnesses display more positive attitudes and behaviours towards patients, including willingness to help [22]. A positive relationship between beliefs and behaviours towards children with ADHD has been demonstrated in teachers with experience teaching children with ADHD [27]. There is, however, very limited literature exploring staff attitudes specifically about CD. Some studies suggest that teachers view children with CD as disruptive and have concerns about the safety of other students during lessons [23–25], indicating negative beliefs. Furthermore, teachers have reported feeling "disgust" and "fear" towards children with CD, implying negative overall affect [24]. In non-teaching professions, attitudes appear more varied. A mixed-methods study using a questionnaire and an interview suggested that nurses and psychologists may hold more positive attitudes towards children with CD than doctors [30]. Inferences from this work are limited, however, due to a lack of statistical appraisal of findings.

Attitudes can be broken down into several components, including the tripartite model, which focuses on affect, beliefs, and behaviour [31]. Affect includes feelings that are aroused after object exposure [31]; cognition refers to beliefs about the object [31], which can be stereotypic (about characteristics, such as "children with CD are disruptive") or symbolic (about the holder's traditions, such as "teaching children with CD is time consuming; [32]) while behaviour consists of actions towards the object or individual [31]. This model has been used to explore attitudes of professionals about ADHD. One study [33] used an adapted version of the Multidimensional Attitudes towards Inclusive Education Scale (MATIES; [32]) and found that psychologists have significantly more knowledge, favourable affect, favourable behaviour, and favourable attitudes than teachers and support assistants. Another study [27] adapted open-ended response measures from Haddock & Zanna [34] and demonstrated that experience elicits differences in knowledge and attitude components. Both the MATIES [32] and open-ended response measures have demonstrated validity when exploring attitude components [32, 35, 36]. To our knowledge, however, no study has used this model to explore attitudes towards CD.

Both knowledge and attitudes can also be impacted by direct and indirect experience with a disorder. Research demonstrates that direct experience of working in general mental health

settings reduces negative attitudes towards patients because unrealistic preconceptions are challenged, such as the fear that psychiatric patients pose a threat and people with diagnoses are out of control [37]. Although evidence relating to CD is limited, qualitative work shows that direct experience of teaching children with disruptive behaviour disorders, including CD, contributes to greater confidence in teachers' knowledge about the disorders and management techniques for these [23, 24]. Similarly, indirect experience with mental health disorders, such as educational training programmes about psychiatric patients, improves attitudes towards patients, because it provides a rationale for their behaviours [38]. One study shows that training through lectures and information leaflets about CD produced significant increases in knowledge and favourable attitudes about CD, compared to a group who received no training [39]. However, research on the impact of experience on knowledge and attitudes towards children with CD appears limited and thus may contribute to the insufficient support received by children with CD.

To address the lack of evidence about knowledge and attitudes towards CD in professionals working with CD, we administered an online questionnaire exploring knowledge, global attitudes, and the tripartite attitude components to four groups of professionals (Psychology Staff, Teaching Staff, Care Staff, and Other Non-Clinical Staff), with varied levels of experience and education or training on CD, who work with young people. Drawing on a limited previous literature, we hypothesised that (i) Psychology Staff would have significantly greater knowledge, favourable global attitudes, and favourable attitude components (tripartite model) than other staff groups; (ii) individuals with direct experience with young people with CD would have significantly greater knowledge, favourable global attitudes, and favourable attitude components than individuals without direct experience, across the entire sample (our design was not powered to perform within-group analyses); and (iii) individuals with indirect experience of CD (training/education) would have greater knowledge, favourable global attitudes, and favourable attitude components than individuals without indirect experience, across occupational groups. To explore the relationship between knowledge, including knowledge subscales (causes, treatments, and characteristics), and attitudes, we conducted secondary exploratory correlational analyses between these variables, across the entire sample.

## Methods

### Participants and procedure

To obtain 80% power, allowing for a detection of a large effect size of 0.8, with a significance threshold of $p < 0.05$, at least 24 participants were required for one-way Analysis of Variance (ANOVA) tests and 52 participants for two-way independent sample t-tests, calculated by G*Power. Participants were required to provide informed consent and to be working with young people aged 10–17 in the UK. This age range was chosen to achieve a balance between maximising sample size and consistency across the clinical population in question, i.e. to include pre-adolescent and adolescent youth but not very young children.

Fifty-nine subjects were recruited from March to May 2021, through convenience and snowball sampling via social media. Advertisements for the study were posted and promoted on the researchers' main and professional accounts on social media (Facebook, LinkedIn, and Twitter), as well as various research and dissertation groups on Facebook. Advertisement posts were also shared by numerous members of the public to reach a wider audience. The occupational groups included psychologists, therapists, and interventions facilitators (referred to henceforth as 'Psychology Staff'; n = 18); teachers and teaching assistants ('Teaching Staff'; n = 19); and nurses and care/support staff ('Care Staff'; n = 15). There were also staff in non-clinical occupations, including sports coaches and managers ('Other Non-Clinical Staff';

n = 6). One doctor, who did not fit into any of these four occupation groups, was excluded, resulting in an overall sample of 58 participants.

Written ethical approval was obtained from the University of Nottingham Faculty of Medical and Health Sciences Ethics Committee (reference FMHS 189–0221). All methods were performed according to UKRI's policy and guidelines. Advertisements were posted on social media sites, including Twitter, Facebook, and LinkedIn. Participants gave written consent through an online consent form. Participants were presented with all materials via an online questionnaire. Upon completion, participants were given the opportunity to provide their email address if they wished to enter a prize draw. Authors had access to email addresses, though they were removed from data analysis following the prize draw. A random number generator chose four prize draw winners after the questionnaire closed, who were contacted via email.

## Design

A cross-sectional design was used, with three independent variables: occupation, direct experience with CD, and indirect experience (education/training) with CD. The dependent variables were all quantitative scale variables based on scores from the questionnaire: total knowledge, global attitude, stereotypic beliefs, symbolic beliefs, affect towards CD, and past behaviour towards CD.

## Materials

An online questionnaire was presented on Online Surveys (www.onlinesurveys.ac.uk), including the information sheet, consent form, questionnaire, prize draw details, and debriefing sheet. The following demographic information was collected: gender, age, where they work with young people, whether the setting is forensic or non-forensic, their main job role, whether their work is full-time/part-time/bank, and years of employment in this role. Education and training were accounted for through asking whether they had completed any university units or training on CD. Experience with CD was measured through asking whether they have worked with a child believed to have CD. The questionnaire contained the sections outlined below, adapted from an ADHD study [27] because, to our knowledge, a questionnaire exploring knowledge and all attitude components in CD does not exist. The original ADHD questionnaire that this study adapted [27] used a shortened version of the Knowledge about Attention Deficit Disorder Questionnaire [40] to optimise rates of inclusion. Both ADHD scales previously demonstrated internal consistency when exploring subscales of knowledge [27, 40].

**Knowledge.** The knowledge measure was adapted from Anderson et al. [27]. New questions were developed, drawing from literature on relevant items, using the original format of 33 questions (see S1 Appendix for the tool, its format, and references). There were three 11-item subscales exploring knowledge of (i) causes, (ii) treatments, and (iii) characteristics. The order of questions for each subscale and true answers remained the same as the original measure by Anderson et al. [27]. Statements included "only biological factors cause conduct disorder"; "antipsychotic drugs can be used to manage symptoms of conduct disorder"; and "language impairments are common amongst children with conduct disorder". Participants were asked to respond with 'true', 'false' or 'don't know' for each item. Correct answers were summed to form total scores from 0 (low knowledge) to 33 (high knowledge). Subscale scores were devised by summing correct answers from each subscale.

**Global attitude.** A vertical, 11-point, attitude thermometer scale determined overall attitudes towards children with CD. Thermometer scales have concurrent validity in samples of

adults when exploring attitudes towards stigmatised groups [35, 41]. Attitudes were rated from 1 (extremely unfavourable) to 100 (extremely favourable).

**Attitude components.**   Attitude components were selected using the Tripartite Model of Attitudes. Separate attitude components (stereotypic beliefs, symbolic beliefs, affect, and past behaviour) were explored using four open-ended questions. The four items were: "please describe what you think the characteristics a child with conduct disorder would have" (stereotypic beliefs), "please say how you think children with conduct disorder affect or may affect your work with them" (symbolic beliefs), "please describe how you feel when thinking about working with children with conduct disorder" (affect), and "please describe how you think you have acted when working with a child with conduct disorder" (past behaviour). Participants wrote up to 12 words or phrases answering each component statement, relevant to CD. Participants transformed their own responses into quantitative data themselves by rating each word or phrase on a 5-point scale ranging from 'very negative' (-2) to 'very positive' (+2). Each participant's four individual scores for each of the four components were formed by calculating a means score of their ratings given to their responses to each statement. This method of interpreting open-ended questions is a valid and reliable technique of eliciting attitudinal responses exploring various components [34, 36, 42]. Cognitive and affective open-ended measures average at 0.75 split-half reliability [34]. Discriminant validity between cognitive and affective component responses has also been established [34].

## Statistical analyses

All analyses were performed using IBM SPSS Statistics version 27. One-way ANOVAs were performed with occupation as an independent variable for each of: total knowledge, global attitude, stereotypic beliefs, symbolic beliefs, affect, and past behaviour. Separate t-tests were conducted for the direct experience variable and the training/education variable with the aforementioned dependent variables.

Exploratory secondary analyses investigated the relationship between knowledge variables (including the knowledge subscales) and attitude variables. Pearson's correlation coefficient tests were performed, followed by three partial correlation tests, controlling for occupation, experience with CD, and training/education separately.

Some data was missing from attitude component questions, so participants' responses were only included for the components they answered to ensure maximum data inclusion. Calculations were carried out on the following number of remaining participants: stereotypic beliefs (n = 50, missing 8 [13.79%]), symbolic beliefs (n = 47, missing 11 [18.97%]), affect (n = 48, missing 10 [17.24%]), and past behaviour (n = 36, missing 22 [37.93%]).

## Results

### Demographic variables

Forty-nine females and nine males participated (see Table 1). The mean age was 32.5 (SD +/- 11.0). Psychology Staff included psychologists, therapists, and interventions facilitators working in environments including schools, secure homes, and the community. Teaching Staff included teachers and teaching assistants working mostly in primary or secondary schools, with one working in a Special Educational Needs school. Care Staff included nurses and care/support staff working in environments including child and adolescent mental health inpatient wards, secure children's homes, and the community. Finally, the Non-Clinical Staff group included three sports coaches/teachers in dance schools and youth centres, mentors at youth centres, and a manager in a young offender institution. It is important to highlight that the

**Table 1. Demographics and details of direct and indirect experience of participant groups.**

| Demographic variables | Psychology Staff (n = 18) | Teaching Staff (n = 19) | Care Staff (n = 15) | Other Non-Clinical Staff (n = 6) | Significance (p) |
|---|---|---|---|---|---|
| **Gender** | | | | | |
| Male | 5 | 2 | 2 | 0 | 0.41 |
| Female | 13 | 17 | 13 | 6 | |
| **Experience working with CD** | | | | | |
| Yes | 13 | 7 | 11 | 2 | <0.001** |
| No | 5 | 12 | 4 | 4 | |
| **Received training/education on CD** | | | | | |
| Yes | 12 | 2 | 3 | 0 | 0.05 |
| No | 6 | 17 | 12 | 6 | |
| **Environment** | | | | | |
| Forensic | 6 | 0 | 8 | 2 | 0.001** |
| Non-forensic | 12 | 19 | 7 | 4 | |
| Age in years: *M (SD)* | 35.0 (8.1) | 30.5 (11.9) | 32.6 (12.2) | 31.5 (13.2) | 0.14 |

Across-group comparisons for gender, experience with CD, training/education about CD, and environment were analysed using Fisher's Exact test. Comparisons for age were analysed using one-way ANOVAs.

sample size is relatively small. It is also predominantly female, however this may be representative of the number of females working with CD in the general population [43].

## Internal consistency of the knowledge of conduct disorder scale

Internal consistency, measured by Cronbach's alpha, was 0.90 for the overall knowledge scale, 0.79 for the causes subscale, 0.73 for the treatment subscale, and 0.75 for the characteristics subscale.

## Effect of occupation on knowledge and attitudes

Table 2 shows mean scores for occupation. There were no significant differences between occupation groups in their total knowledge ($F_{(3, 22.47)}$ = 1.62, $\eta^2$ = 0.08, p = 0.21). See Fig 1 for knowledge scores grouped by occupation. There were, however, significant differences between occupation groups in global attitudes towards CD ($F_{(3, 54)}$ = 4.36, $\eta^2$ = 0.20, p = 0.01). See Fig 2 for global attitude scores grouped by occupation. Post-hoc between-group analyses revealed that Psychology Staff had significantly more favourable global attitudes than Teaching

**Table 2. Effect of occupation on knowledge and attitudes.**

| | Psychology Staff | | | Teaching Staff | | | Care Staff | | | Other Non-Clinical Staff | | | $\eta^2$ |
|---|---|---|---|---|---|---|---|---|---|---|---|---|---|
| | M | SD | n | M | SD | n | M | SD | n | M | SD | n | |
| **Total knowledge** | 20.7 | 4.7 | 18 | 15.5 | 9.7 | 19 | 17.8 | 7.2 | 15 | 18.7 | 3.8 | 6 | .08 |
| **Global attitude** | 75.5 | 19.8 | 18 | 50.8 | 26.1 | 19 | 65.9 | 16.7 | 15 | 58.3 | 16.9 | 6 | .20** |
| *Attitude components* | | | | | | | | | | | | | |
| **Stereotypic beliefs** | -0.4 | 0.8 | 17 | -1.0 | 0.6 | 16 | -0.9 | 0.5 | 12 | -0.8 | 1.2 | 5 | .14 |
| **Symbolic beliefs** | 0.2 | 1.2 | 16 | -0.1 | 0.8 | 15 | -0.3 | 1.3 | 11 | -0.4 | 1.1 | 5 | .18* |
| **Affect** | 0.6 | 1.1 | 16 | -0.1 | 1.0 | 16 | 0.0 | 1.2 | 11 | -0.4 | 0.9 | 5 | .10 |
| **Past behaviour** | 1.1 | 0.9 | 13 | 0.8 | 0.6 | 11 | 0.4 | 1.0 | 10 | 1.0 | 1.4 | 2 | .11 |

* p < .05

** p < .01

*** p < .001.

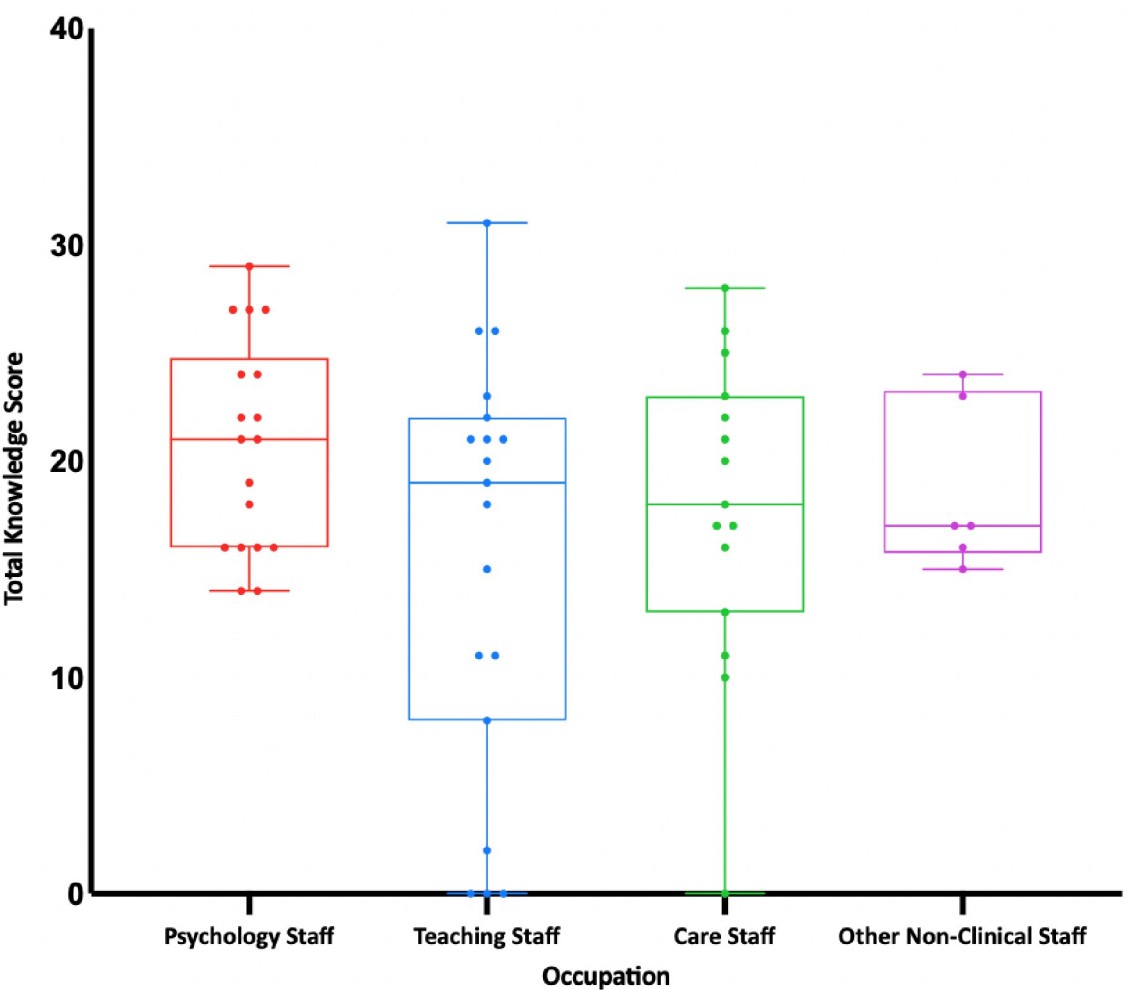

**Fig 1. Total knowledge scores grouped by occupation.**

Staff ($F = 0.49$, p = 0.01). No other group differences were significant. There were also significant differences between occupation groups in symbolic beliefs ($F_{(3, 43)} = 3.24$, $\eta^2 = 0.18$, p = 0.03). Post-hoc between-group analyses revealed that Psychology Staff had significantly more favourable symbolic beliefs than Teaching Staff ($F = 0.57$, p = 0.02). No other group differences were significant.

## Effect of direct experience on knowledge and attitudes

Table 3 shows mean scores for individuals with and without direct experience working with young people with CD across all occupational groups. Individuals with direct experience displayed significantly higher knowledge ($t_{(34.80)} = 3.41$, d = 0.97, p = 0.002) compared to individuals without experience. Global attitudes of individuals with direct experience were significantly more favourable than those without experience ($t_{(56)} = 4.21$, d = 1.12, p < 0.001). However, no significant differences were found for stereotypic beliefs ($t_{(48)} = 0.74$, d = 0.21, p = 0.47), symbolic beliefs ($t_{(45)} = 1.74$, d = 0.51, p = 0.09), affect ($t_{(46)} = 1.89$, d = 0.55, p = 0.07), or past behaviour ($t_{(34)} = .34$, d = 0.12, p = 0.73). There were insufficient numbers in each occupational group to conduct within-group analyses of the effect of direct experience.

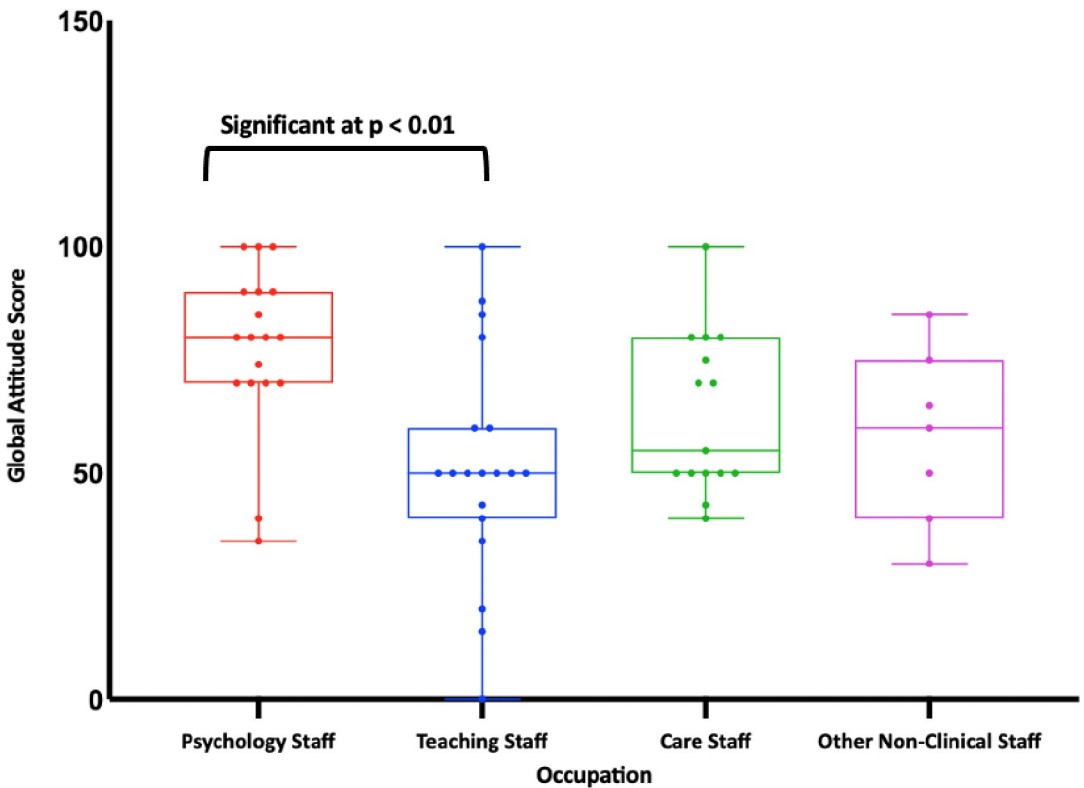

**Fig 2. Global attitude scores grouped by occupation.**

### Effect of indirect experience (training and education) on knowledge and attitudes

Table 3 also shows mean scores for individuals with and without training/education across all occupational groups. Individuals who had received training/education showed significantly higher knowledge ($t_{(56)} = 2.99$, d = 0.86, p = 0.004) compared to individuals who had not.

**Table 3. Differences in attitudes in individuals with and without direct experience and indirect experience (training/education).**

| | Direct experience | | | | | Indirect experience (training/education) | | | | |
|---|---|---|---|---|---|---|---|---|---|---|
| | With | | Without | | d | With | | Without | | d |
| | M | SD | M | SD | | M | SD | M | SD | |
| **Total knowledge** | 21.0 | 4.7 | 17.2 | 7.4 | 0.97** | 22.4 | 5.1 | 18.5 | 6.0 | 0.86** |
| **Global attitude** | 69.8 | 17.9 | 57.5 | 35.3 | 1.12*** | 72.7 | 18.6 | 62.6 | 21.7 | 0.68* |
| **Stereotypic beliefs** | -0.7 | 0.7 | -0.5 | 1.0 | 0.21 | -0.7 | 0.9 | -0.6 | 0.7 | 0.13 |
| **Symbolic beliefs** | -0.1 | 1.1 | -0.4 | 1.3 | 0.51 | 0.0 | 1.2 | -0.4 | 1.2 | 0.59 |
| **Affect** | 0.4 | 1.1 | 0.2 | 0.8 | 0.55 | 0.6 | 0.9 | 0.2 | 1.1 | 0.50 |
| **Past behaviour** | 0.8 | 0.9 | 0.9 | 0.9 | 0.12 | 0.9 | 0.9 | 0.8 | 0.9 | 0.07 |

* p < .05

** p < .01

*** p < .001.

Global attitudes were significantly more favourable in those who had received training/education compared to those who had not ($t_{(56)}$ = 2.35, d = 0.68, p = 0.02). There were no significant group differences in stereotypic beliefs ($t_{(48)}$ = 0.40, d = 0.13, p = 0.69), symbolic beliefs ($t_{(45)}$ = 1.81, d = 0.59, p = 0.08), affect ($t_{(46)}$ = 1.52, d = 0.50, p = 0.13), or past behaviour ($t_{(34)}$ = 0.19, d = 0.07, p = 0.85). There were insufficient numbers in each occupational group to conduct within-group analyses of the effect of training/education.

### Relationships between knowledge and attitude measures

Table 4 shows the bivariate correlations for the relationship between knowledge and attitudes across all occupational groups. The following factors had significant, positive relationships: global attitudes with total knowledge and all knowledge subscales, affect with knowledge of causes and knowledge of characteristics, and past behaviour with knowledge of causes, knowledge of characteristics, and total knowledge.

The partial correlations are presented in Tables 5 and 6. In summary, when controlling separately for both direct and indirect (training/education) experience, global attitude was no longer significantly correlated with knowledge and the knowledge subscales. Also, the relationships between both affect and past behaviour with knowledge of characteristics was no longer significant. However, the relationships between both affect and past behaviour with knowledge of causes, and past behaviour with total knowledge, remained significant.

## Discussion

We examined knowledge and attitudes about CD in professionals working with young people with the condition. There were three key findings. Firstly, Psychology Staff had significantly more favourable global attitudes and symbolic beliefs than Teaching Staff (though not other staff groups, therefore only partly supporting our hypothesis). Secondly, also partially supporting our hypothesis, those with direct experience working with young people with CD had significantly greater total knowledge and global attitudes than those without, though no significant differences were found in any attitude components. Thirdly, those with indirect experience (training or education) about CD had significantly greater objective knowledge and global attitudes than those without, though there were no significant differences in any attitude subcomponents.

### Effect of occupation on knowledge and attitudes

Unexpectedly, there were no significant differences in knowledge between occupation groups. Previous studies have demonstrated that teachers lack knowledge about CD [23–25]. It is notable that Psychology Staff did have higher mean scores for knowledge than all other groups in

**Table 4. Bivariate correlations between knowledge and attitude measures.**

| Type of knowledge about CD | Global attitude | Stereotypic beliefs | Symbolic beliefs | Affect | Past behaviour |
|---|---|---|---|---|---|
| **Causes** | 0.30* | 0.14 | 0.25 | 0.20* | 0.30*** |
| **Treatments** | 0.30* | 0.00 | 0.00 | 0.14 | 0.12 |
| **Characteristics** | 0.32* | 0.06 | 0.26 | 0.36* | 0.54** |
| **Total** | **0.34**** | **0.08** | **0.20** | **0.27** | **0.38*** |

* p < .05

** p < .01

*** p < .001, two-tailed.

**Table 5. Partial correlations between knowledge and attitude measures after controlling for direct experience with CD.**

| Type of knowledge about CD | Global attitude | Stereotypic beliefs | Symbolic beliefs | Affect | Past behaviour |
|---|---|---|---|---|---|
| Causes | 0.17 | 0.02 | 0.20 | 0.30* | 0.59*** |
| Treatments | 0.17 | 0.12 | 0.22 | 0.15 | 0.31 |
| Characteristics | 0.08 | -0.06 | -0.12 | 0.02 | 0.15 |
| Total | **0.16** | **0.04** | **0.13** | **0.18** | **0.42*** |

* p < .05
** p < .01
*** p < .001, two-tailed.

our sample, and it is possible that a larger sample size may have elicited significant differences between groups. However, the relatively low scores across all professional groups (62.7% for Psychology Staff, 47% for Teaching Staff, 53.9% for Care Staff, and 56.7% for Other Non-Clinical Staff) suggests that there may be a lack of specific training in CD for staff working in child and adolescent services, including psychologists.

The finding that global attitudes were significantly more favourable in Psychology Staff than Teaching Staff is in keeping with limited previous research focusing on attitudes towards children with CD, which demonstrates that psychologists, nurses, and therapists held more positive attitudes than doctors [30], and teachers appear to hold mostly negative views [23, 24]. It is also broadly in keeping with research in ADHD, demonstrating that psychologists had more positive attitudes than teachers [33]. One potential explanation is more favourable pre-existing attitudes, with those interested in a career in psychology tending to be more understanding of complex emotional and behavioural problems. This, however, is not supported by existing research and without data on baseline attitudes, our study cannot clarify this further. An additional, or alternative, contributory factor is differing occupation-based experiences and responsibilities. Teachers promote learning skills in a safe environment, with a responsibility for maintaining the best classroom environment across groups of students, and must contend with typical CD behaviours as disruptive towards learning and other students. When signs of behavioural disturbance or mental health issues emerge, they refer children elsewhere for assessments [13]. In contrast, psychology staff are responsible for assessing CD and providing psychosocial interventions at an individual level [9, 44]. Such therapeutic encounters may allow for an emerging understanding and empathy about the condition, in a way that fraught experiences in a classroom environment may not. Similar findings have been observed in patients with schizophrenia. For instance, one study showed that public health nurses were more socially accepting of patients than psychiatric nurses due to potentially seeing patients live in socially responsible conditions; however, they demonstrated both more positive affect and higher social acceptance than non-care workers, potentially because

**Table 6. Partial correlations between knowledge and attitude measures after controlling for indirect experience (training/education).**

| Type of knowledge about CD | Global attitude | Stereotypic beliefs | Symbolic beliefs | Affect | Past behaviour |
|---|---|---|---|---|---|
| Causes | 0.25 | 0.05 | 0.22 | 0.33* | 0.54*** |
| Treatments | 0.21 | 0.13 | 0.18 | 0.13 | 0.31 |
| Characteristics | 0.23 | -0.02 | -0.06 | 0.09 | 0.11 |
| Total | **0.26** | **0.06** | **0.13** | **0.21** | **0.39*** |

* p < .05
** p < .01
*** p < .001, two-tailed.

these staff had more contact and greater educational opportunities [45]. Although we found no significant differences between Teaching Staff and other groups, the overall ('global') attitude of Teaching Staff was the lowest of the four groups, which suggests that teachers may have particular difficulties in understanding or adapting to the disorder and may be faced with particular challenges. This is an important consideration in developing support structures for teachers, particularly in challenging environments. Teachers may need more support when teaching children with CD, which could be informed by psychologists.

Differences between occupational groups were also observed in symbolic beliefs, which represent staff values about how individuals with CD impact their working process (for example, for teachers, beliefs about how teaching a class may be impacted if one student has CD). Individuals may deduce beliefs from experiences at work because experiences play a critical role in belief formation [46]. Therefore, differences between occupations may simply arise because the item specifically focuses on values towards their occupation, supporting the view that occupation-based experiences may impact global attitudes. Conversely, the lack of differences between occupation groups in their stereotypic beliefs, affect, and past behaviour may be because these items focused less on their occupation and more on their evaluation towards the individual with CD, perhaps explaining the lack of significant difference between occupation groups. This is supported by previous research demonstrating that teachers report many concerns when teaching children with CD [23, 24], but psychologists and therapists do not believe children with CD are especially challenging to work with [30].

## Effect of direct and indirect experience on knowledge and attitudes

Direct experience was associated with greater knowledge, across the entire sample. Our findings support previous studies, demonstrating that increased experience with CD [23, 24] and ADHD [27, 47] led to improvements in knowledge of each disorder. Knowledge can be shaped through observation, reflection, conceptualisation, and adaptation from experiences, a process called experiential learning [48]. Therefore, individuals who have direct experience have likely undergone this process and may have improved their knowledge in this way. Furthermore, negative preconceptions are usually challenged through gaining direct experience with mental health disorders [37]. This should be considered in training programmes for all professionals likely to come into contact with CD. Direct encounters through work experience and placements may be a valuable means of optimising learning in these important professional groups.

As predicted, knowledge was also higher in individuals with training/education than those without. Previous research supports that training/education about CD improves knowledge about CD [23–25, 39]. Specifically, one study suggests that training individuals about various factors (including risk factors, symptoms, and treatment) contributed to an overall increase in knowledge of CD [39]. This reinforces the idea that targeting specific aspects is essential when exploring knowledge of CD. However, this research on CD training only explored a combination of a lecture, two videos, and an information leaflet. Printed materials [49] and lectures [50] have been shown to have minimal effects on clinical knowledge, whereas interactive groups [50] and courses with practical components [51–53] are more effective. Therefore, controlling for types of training methods would provide greater insight into the impact it has on knowledge. This may also have further implications for knowledge about CD, when techniques such as video call lectures are increasingly used following the COVID-19 pandemic.

Also as predicted, global attitudes were more favourable in individuals with direct experience of CD and training/education compared to individuals without experience and training/education, respectively. This may result from empathic responses to individuals with CD, where professionals may explain behaviours as a result of a diagnosis of CD [54]. Furthermore,

those who seek out training in CD may have pre-existing favourable attitudes to CD. Supporting this, previous findings suggest a relationship between having more empathy and a greater understanding of mental health issues [54]. The relative contribution of these factors should be explored in future research, controlling for pre-existing attitudes.

Contrary to our hypothesis, neither training/education nor experience working with children with CD had an effect on specific the attitude components within the tripartite model. Theorists suggest that beliefs are particularly resistant to change, more-so than factual knowledge [55, 56], perhaps explaining why there were no differences in symbolic or stereotypic beliefs. This has also been observed in ADHD research [27]. Therefore, their experience may not be sufficient enough to change beliefs.

## Exploratory analysis: Relationships between knowledge and attitudes

Our secondary exploratory analysis revealed that all aspects of knowledge were positively correlated with global attitudes. This is in keeping with past findings that demonstrate that individuals with greater knowledge have more positive attitudes about CD [39] and other disorders, such as ADHD [27, 33, 57]. The directionality of this relationship warrants exploration in future research studies.

Interestingly, knowledge about causes of CD was also positively correlated with affect and past behaviour as well as global attitudes. This suggests that a greater understanding of the aetiology of CD may have beneficial effects on approaches to individuals with the condition. Recent research has increasingly pointed to a considerable genetic and neurobiological contribution to CD [58–60], particularly for CD with CU traits [61]. Knowledge of these factors, however, may hypothetically influence professionals in two ways. On one hand, understanding these biological contributions may make them more sympathetic toward individuals with CD, given their predilection to developing the disorder and its sequelae. On the other hand, it may cause professionals to be guarded about CD, and develop a therapeutic hopelessness about outcomes for the disorder, seeing affected individuals as beyond the influence of usual psychosocial interventions. This has parallels in the field of 'Neurolaw', whereby presentation of neurobiological evidence of potential causation to juries and judges has been shown to be both an aggravating and a mitigating factor in sentencing [62, 63]. In our study, it would appear that the former is the case; greater knowledge about causes leads to a more sympathetic attitude towards the condition. This has positive implications for young people suffering with CD and their families and puts an onus on clinical services to help educate staff and the general public.

The relationship between all knowledge scales and global attitudes, however, was no longer significant after controlling for occupation, experience with CD, and training/education. Previous research demonstrates that a relationship between knowledge and attitudes towards ADHD exists for teachers with experience, but not for those without [27]. Our sample was insufficiently powered to carry our subgroup analyses of the interrelationship, however future work would benefit from further exploration of this.

## Strengths and limitations

This is the first study to examine the impact of occupation and direct and indirect experience on knowledge and attitudes towards children with CD. We identified a sample of relevant professionals and used an adapted version of validated and reliable measures, providing the first evidence for internal consistency of this adapted version. Our study, however, had several limitations. Firstly, smaller effects may have been missed and the scope of subgroup analyses was limited. Despite this study being promoted and widely circulated online, uptake and completion was limited to a relatively small sample of 58. There are several possible reasons for this, two of

which warrant particular mention. Firstly, it may be that many staff working with CD are unaware or unclear about of the concept of CD, because it is rarely diagnosed [14], and so the advert would not have captured their attention. This highlights the importance of psychoeducation about CD for staff within educational and healthcare settings for young people. Secondly, CD is a contentious issue. Some argue that CD does not exist and the diagnosis is harmful, whereas others believe a diagnosis can help highlight treatment needs [64]. Even within groups of staff who are aware of the CD concept, some individuals who work with youths with CD may be reluctant to validate the concept by engaging in research using this terminology. Frank and open discussions about the quality of evidence for CD and the potential ethical and practical implications of use of this diagnosis are important for researchers in this field to embrace.

Secondly, there were significant within-group differences in training/education within occupation and direct experience groups. This meant that exploring the effect of occupation on knowledge and attitudes may have been less meaningful than more homogenous subgroup samples would have allowed.

Thirdly, individuals with direct experience were more likely to work in a forensic setting, perhaps since the prevalence of CD is greater in forensic settings [65]. This limits the generalisability of our findings to non-forensic settings, such as schools. Furthermore, the sample was predominantly female, and the females in the sample had more training/education than males. This may well be representative of the make-up of professionals working with CD in the general population [43]. However, sex has been shown to impact attitudes, whereby females hold more positive views than males towards individuals who have committed sexual offences [66] and students with disorders including CD and anxiety [67].

## Conclusions

Our study contributes to the evidence base suggesting psychology staff have more favourable attitudes to conduct disorder than teachers, though the reasons for this remain unclear. Both direct and indirect experience were associated with greater knowledge and more favourable attitudes, perhaps because experience challenges negative preconceptions about CD. Future research will benefit from a longitudinal approach exploring the directionality of the relationship between knowledge and attitudes and larger sample sizes to more fully elucidate potential differences between occupational groups. Teachers in particular may require extra support in dealing with youths with this challenging condition.

## Supporting information

**S1 Appendix. List of items on objective knowledge of conduct disorder scale, corresponding citations, and scoring instructions.** The items that correspond to each of the subscales (characteristics, treatments, and causes) can be seen. Also, it states which items are true or false. Finally, Table 7 in S1 Appendix demonstrates the items and their corresponding citations, followed by a list of references.
(DOCX)

## Acknowledgments

We express gratitude to all staff who took time to participate in this study.

## Author Contributions

**Conceptualization:** Chloe Pinchess, John Tully.

**Formal analysis:** Chloe Pinchess, John Tully.

**Investigation:** Chloe Pinchess, John Tully.

**Methodology:** Chloe Pinchess, John Tully.

**Supervision:** John Tully.

**Writing – original draft:** Chloe Pinchess.

**Writing – review & editing:** Ruth Pauli, John Tully.

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
