## [Decision Letter · Decision Letter 0]

20 Jul 2023

PONE-D-23-06904Knowledge and attitudes about conduct disorder of professionals working with young people: the influence of occupation and direct and indirect experiencePLOS ONE

Dear Dr. Pinchess,

Thank you for submitting your manuscript to PLOS ONE. After careful consideration, we feel that it has merit but does not fully meet PLOS ONE’s publication criteria as it currently stands. Therefore, we invite you to submit a revised version of the manuscript that addresses the points raised during the review process.

Thank you for your submission. I have reviewed your manuscript as well as the comments of the reviewer. In addition to their comments, I also have the following comments:

-Please retain information about the prize draw and confidentiality of emails in the revision (disregard this one comment from the reviewer).

-In addition to including the eligibility criteria (as requested by Reviewer 1), please also justify how the 10-17 age range for clients/patients was chosen.

-Please include information on the percentage and number of participants with missing data, how missing data were handled, and why the method of addressing missing data was chosen

-Please clarify if the knowledge measure was developed for this study or is pre-existing. Please explain how the measure was developed and checked for accuracy and include the measure as an appendix if newly developed.

We look forward to receiving your revised manuscript.

Kind regards,

Emily Lund

Academic Editor

PLOS ONE

Journal Requirements:

Additional Editor Comments:

Thank you for your submission. I have reviewed your manuscript as well as the comments of the reviewer. In addition to their comments, I also have the following comments:

-Please retain information about the prize draw and confidentiality of emails in the revision.

-In addition to including the eligibility criteria (as requested by Reviewer 1), please also justify how the 10-17 age range for clients/patients was chosen.

-Please include information on the percentage and number of participants with missing data, how missing data were handled, and why the method of addressing missing data was chosen

-Please clarify if the knowledge measure was developed for this study or is pre-existing. Please explain how the measure was developed and checked for accuracy and include the measure as an appendix if newly developed.

Reviewers' comments:

Reviewer's Responses to Questions

**Comments to the Author**

1. Is the manuscript technically sound, and do the data support the conclusions?

Reviewer #1: Yes

2. Has the statistical analysis been performed appropriately and rigorously? 

Reviewer #1: Yes

3. Have the authors made all data underlying the findings in their manuscript fully available?

Reviewer #1: Yes

4. Is the manuscript presented in an intelligible fashion and written in standard English?

Reviewer #1: Yes

5. Review Comments to the Author

Reviewer #1: Introduction - Overview of the topic adequate

Methods:

1. The study is incorrectly described as a between subjects design, but the statistical analysis used is appropriate for a cross-sectional survey design

2. Information about the email addresses and prize draw isn't necessary for this paper

3. Eligibility criteria for participants should be made more explicit

4. How was the study size arrived at? If a sample size calculation was performed, please state.

5. Please provide more detail about recruitment methods - clarify if recruiting through specific groups and networks as this can be relevant to generalisability of findings

6. Please provide more information about the survey. For example, what demographic information was collected? What information was collected about experience? Detail any pilot testing that was done

7. Please provide more information about how open-ended responses were analysed, that is, how did you group the responses? Only the scoring information is provided.

Results:

1. Provide information on representativeness of the sample - this is mentioned in discussion but should be stated clearly in your results.

Discussion

1. In the limitations it says that the study was powered to detect large to moderate effects - please correct this language as no power analysis is provided in your methods section, and power analysis may be incorrect based on the study design stated in the methods.

2. Sample size is a major limitation of this study - can you provide details for why the sample was so small? the population being drawn from is over 1 million (teachers in UK alone are about 500K), so having only 58 participants in this study bears some introspection. Detail about recruitment methods will also help understand the generalisability - if recruiting through networks with more CD experience/interest this is very relevant.

3. Please change biological sex to gender or omit the word 'biological'.

6. PLOS authors have the option to publish the peer review history of their article (what does this mean?). If published, this will include your full peer review and any attached files.

Reviewer #1: No

---

## [Author Response · Author response to Decision Letter 0]

31 Aug 2023

Dear Reviewers,

Thank you for your comments. I have addressed them in the following documents:

- Revised Cover Letter

- Manuscript

- Manuscript with Track Changes 

- Letter to Reviewers

- S1 Appendix.

Best wishes,

Chloe

---

## [Editor Report · Decision Letter 1]

18 Sep 2023

Knowledge and attitudes about conduct disorder of professionals working with young people: the influence of occupation and direct and indirect experience

PONE-D-23-06904R1

Dear Dr. Pinchess,

We’re pleased to inform you that your manuscript has been judged scientifically suitable for publication and will be formally accepted for publication once it meets all outstanding technical requirements.

Kind regards,

Emily Lund

Academic Editor

PLOS ONE

Additional Editor Comments (optional):

Thank you for your clear and thorough responses to previous comments and requests from the reviewer and editorial team.
---

## [Editor Report · Acceptance letter]

21 Sep 2023

PONE-D-23-06904R1 

Knowledge and attitudes about conduct disorder of professionals working with young people: the influence of occupation and direct and indirect experience 

Dear Dr. Pinchess:

I'm pleased to inform you that your manuscript has been deemed suitable for publication in PLOS ONE. Congratulations! Your manuscript is now with our production department. 

Kind regards, 

on behalf of

Dr. Emily Lund 

Academic Editor

PLOS ONE